# Designing Polaritonic Integrated Circuits for Quantum Processing

**Mathias Van Regemortel**[*], **Wolfger Peelaers**[*], **Thomas Van Vaerenbergh**
Hewlett Packard Labs
HPE Belgium
B-1831 Diegem, Belgium
{mathias.van-regemortel, wolfger.peelaers, thomas.van-vaerenbergh}@hpe.com

## Abstract

We propose photonic integrated circuits augmented with a $\chi^{(3)}$ nonlinearity – e.g., a semiconductor exciton-polariton nonlinearity – to accomplish two fundamental tasks in quantum processing: quantum state tomography and quantum state generation. We demonstrate in simulations that the design of the circuit can be optimized to great effect and showcase the efficacy of the optimized nonlinear circuits for the quantum machine learning tasks of i) fully identifying a family of emblematic quantum states and ii) stabilizing an accurate train of single photons.

## 1 Introduction

Advancements in quantum computing heavily rely on the manipulation, stabilization, and detection of quantum states [1, 2]. Ever since its conception, the generation and characterization of non-classical quantum states of light have been at

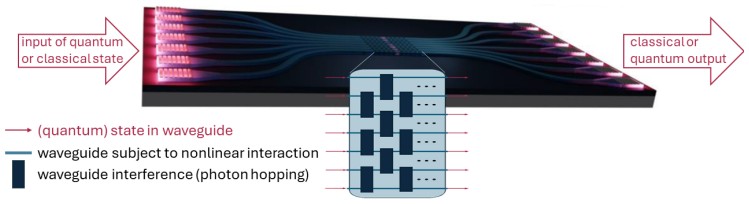

Figure 1: Schematic depiction of our photonic integrated chip.

the heart of quantum optics [3], thus paving the way towards large-scale optical quantum computing [4, 5]. While in the classical approach to optical computing [6, 7] information is captured solely in the relatively easily measurable and modulatable amplitude and/or phase of different components of light, a quantum state of light is fully determined by (i) full-counting photon statistics and (ii) the relative phase of each photon number outcome – features of fundamental importance for quantifying quantum entanglement and state superposition [8].

Strongly coupled matter-light systems, such as exciton-polaritons [9, 10], have attracted growing interest in the quantum-computing community. Thanks to the flexible state manipulation provided by the incident laser field and direct access to the emitted light, experiments with exciton-polaritons have accomplished a plethora of impressive results – e.g., the creation of spontaneous long-range coherence (Bose-Einstein condensation) [11, 12], realization of a superfluid flow [13, 14], or the stabilization of an optical topologically protected insulator [15, 16]. It is, therefore, not surprising that recently these platforms have also been proposed for optical quantum computing [17–19].

In this paper, we introduce a photonic integrated circuit (PIC), built upon a platform of nonlinear photonics, that creates a novel pathway for achieving universal quantum state generation and characterization. See Fig. 1 for a schematic depiction.

---

[*]Equal contribution.

38th Second Workshop on Machine Learning with New Compute Paradigms at NeurIPS 2024(MLNCP 2024).

**Related work.**   Recently, a series of papers showcased the potential of quantum reservoir computing for the tasks of state tomography – an all-encompassing characterization of a photonic quantum state [20–23] – and state generation [24]. Although the presented results were innovative and notable from a conceptional point of view, limited efforts were dedicated to the actual implementation, the design, or the potential deployment in state-of-the-art experimental settings. The primary goal of this paper is to do precisely that.

The realization of polaritonics [9] in integrated photonics [25] served as a motivational cornerstone for this work. We demonstrate a feasible implementation of both quantum tomography and quantum state generation using polaritonic PICs. Moreover, while reservoir computing is, by definition, a non-tunable technique to enlarge the feature space, we apply machine learning training methods for optical quantum circuit design to shape an optimal feature space for the task at hand.

**Our contributions**   can be summarized as follows:

1. we showcase the potential of integrated nonlinear photonics to perform the quantum machine learning tasks of quantum state tomography and quantum state generation

2. we provide evidence that one can efficiently and to great effect optimize the design of the polaritonic PIC for tomographical tasks

3. we show the efficacy of our polaritonic device for the task of photonic state generation. Specifically, we focus on the technologically relevant case of generating highly accurate trains of single photons, at maximal emission rate.

## 2   Polaritonic Photonic Integrated Circuit Optimization

**On-chip quantum dynamics.**   The full quantum dynamics, describing photons traversing the chip of interconnected nonlinear waveguides illustrated in Fig. 1, is governed by the interplay of energy-conserving Hamiltonian dynamics and dissipation in the form of, primarily, polariton losses. Under the *Born-Markov* approximation, this is fully captured by the Lindblad master equation for the photonic density matrix – see, e.g., Ref. [26]. Using the paraxial approximation, mapping the spatial axis of polariton propagation onto time $t$ (see Ref. [27] and Appendix B), we find,

$$\partial_t \rho = -i\big[H(t), \rho\big] + \gamma \sum_i \left( a_i \rho a_i^\dagger - \frac{1}{2}\big(a_i^\dagger a_i \rho + \rho a_i^\dagger a_i\big)\right),\tag{1}$$

where we set Planck's constant $\hbar$ and the polariton group velocity $v_g$ to one – see also Appendix A. The first term of equation (1) describes the coherent dynamics, defined by the Hamiltonian,

$$H(t) = -\Delta \sum_{i=1}^{L} a_i^\dagger a_i - \sum_{i,j=1}^{L} \big(J_{ij}(t) a_i^\dagger a_j + J_{ij}^*(t) a_i a_j^\dagger\big) + \frac{U}{2}\sum_{i=1}^{L} a_i^\dagger a_i^\dagger a_i a_i,\tag{2}$$

and the second term the photonic losses. Here, $a_i$ and $a_i^\dagger$ are the photon creation and annihilation operators, respectively, which create or remove a photonic quantum from a quantum state $|\psi\rangle$, and $L$ is the number of waveguides. Let us clarify the parameters that were used in Eqs. (1) and (2). Some of them are semiconductor material properties and are, therefore, not easily accessible for tuning. These include the detuning $\Delta$ of the incoming light with respect to the exact polariton resonance – in what follows, we assume exact resonance, so $\Delta = 0$; the polariton loss rate $\gamma$; and the strength of the nonlinear photon interaction $U$ that is mediated by excitons, which are effective bosonic quasi-particles arising from semiconductor excitations.

Of fundamental importance is the dimensionless ratio $U/\gamma$, quantifying the effective nonlinearity experienced by the photon field. While in traditional microcavity platforms a modest optical non-linearity has been reported [28], it has been shown that much larger effects can be engineered in integrated waveguide designs [25].

Besides the fixed properties of the material stack $\Delta$, $U$, and $\gamma$, the set of parameters $J_{ij}(t)$ in (2) remains directly accessible for the purpose of optimizing the circuit design. These parameters describe the graph matrix of waveguide couplings, as fabricated using directional couplers (DCs) or multi-mode interferometers (MMIs). We choose to keep the matrices $J_{ij}$ real, configured in a brick-like pattern of couplings, see Fig. 1.

In this work, it is precisely the flexibility in the design of the PIC, parametrized by $J_{ij}(t)$ in (2), that we exploit to optimize the performance of the chip on two exemplary tasks: (i) the quantum tomography of incoming Gaussian squeezed states of light and (ii) the emission of a single-photon train at maximal emission rate.

**Quantum simulation.** In principle, full exact quantum simulation of the dynamics governed by Eqs. (1) and (2) can be achieved using the toolbox of linear algebra, by representing quantum states $|\psi\rangle$ as vectors and the operators as matrices. Noteworthy, even though the method is, in principle, exact, the vector space dimension grows exponentially as $D = N^L$, for $L$ waveguides carrying photon states up to occupation number $N$. This exponential growth is well-known to constitute the fundamental hardness of classically simulating quantum mechanics. Also worth noting is that for photonic systems, the occupation number can be any positive integer. That's why we rely on computationally more efficient techniques with manageable memory overhead, to wit, a Gaussian variational approximation and the method of Monte Carlo unravelings of the master equation [29]. For details on these simulation techniques we refer the reader to the supplementary material, see Appendices B and C.

**Quantum state tomography and generation.** For quantum state tomography, the photonic integrated circuit establishes a quantum extreme learning machine [30]. Recall that a classical extreme learning machine pre-processes data via a memoryless (i.e., non-recurrent) non-linear map, while the final output is obtained via a trainable linear layer. In its quantum version, the preprocessing of the input data – in our case a quantum state of light – is governed by a quantum-mechanical time-evolution, followed by the quantum measurement of a collection of observable quantities. In our realization, the quantum dynamics is governed by the above-described effects, while we measure the expected photon count at the end of each waveguide.

For quantum state generation, the PIC receives standard laser beams as its input, which are subsequently intertwined into an entangled multi-mode quantum output state, thanks to the interplay of waveguide interference and the photonic nonlinearity. This quantum state is next sent through a traditional (hence, fully linear) mesh of tunable phase-shifters and DCs which performs an entangling mixture governed by an $L \times L$ unitary matrix $V$ and whose output is the desired quantum state.

**Quantum circuit optimization.** For tomographical tasks, going beyond the extreme learning machine paradigm, we optimize the circuit via its coupling coefficients $J_{ij}(t)$, which are easily adjustable during the design phase of the chip. As these coefficients govern the quantum dynamics as in (2), they appear as parameters in the ordinary differential equations (ODEs) describing the time evolution of the quantum state. Choosing a gradient descent approach to perform the optimization, we can rely on the adjoint state (or sensitivity) method to find the updates to the ODE parameters. Within the machine learning community, this approach was reintroduced in [31], and we leverage the associated PyTorch package. Note that the linear layer is always fitted perfectly optimally given the measurements of the occupation numbers (output intensities).

Fig. 2 shows the setup of state generation focused on the task of single-photon generation. We optimize the parameters of the unitary mixing, which we parameterize as $V(\vec{\theta}) := \exp\{i\mathcal{G}(\vec{\theta})\}$. Here, $\mathcal{G}$ is an $L \times L$ Hermitian matrix, representing the inter-waveguide couplings that we want to optimize to obtain stable high-probability single-photon outcomes – this is perfectly similar to the optimization of the coefficients $J_{ij}(t)$ for tomography. For this usecase, we choose to keep the nonlinear PIC random and fixed. The rationale is that we ultimately aim for a device that is universal in its quantum state generation capabilities. There is, however, no obstruction to designing a device tailored to generating a specific state by leveraging the techniques of the previous paragraph.

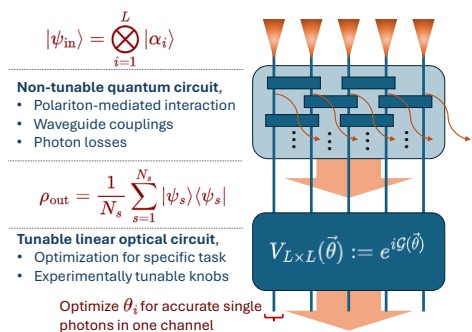

Figure 2: The quantum state generation setup focused on the task of single-photon generation.

# 3 Results

In this section, we present our results, based on numerical simulations, on optimizing the tasks of quantum tomography and state generation. See Appendix E for the simulation details.

## 3.1 Gaussian quantum state characterization

Gaussian quantum states are multi-mode optical states characterized by the property that number statistics – as expressed by expectation values of strings of creation and annihilation operators – are governed by Gaussian statistics. Amongst this class of states, we focus on single- and two-mode thermal squeezed states, often abbreviated as SMTSS and TMTSS respectively. As squeezed versions of thermal states, they are specified by three real parameters: the thermal average number of photons $\bar{n}$ in each mode and the complex squeezing parameter $\xi = re^{i\theta}$. These parameters fully determine the covariance matrix of the above-mentioned Gaussian statistics. We do not displace the state, i.e., we keep the distribution's first moments zero. In formulae, the tomographical information of an SMTSS can be summarized by the following expectation values [3]:

$$\langle a \rangle = 0 , \qquad n = \langle a^\dagger a \rangle = \left( \bar{n} + \frac{1}{2} \right) \cosh(2r) - \frac{1}{2} , \qquad c = \langle aa \rangle = - \left( \bar{n} + \frac{1}{2} \right) \sinh(2r) e^{i\theta} .$$
(3)

For two-mode thermal squeezed states similar expressions determine the equal diagonal elements of $\mathcal{N}_{ij} = \langle a_i^\dagger a_j \rangle$, while the off-diagonal entries are zero, and the equal off-diagonal elements of $\mathcal{C}_{ij} = \langle a_i a_j \rangle$, whose diagonal is zero. Naturally, the indices $i, j$ run over the values 1 and 2. We implement the quantum state characterization of thermal squeezed states as the task of determining $n$ and the real and imaginary parts of $c$.

Concretely, we inject the single/two-mode thermal squeezed state into one/two input waveguides of the chip of Fig. 1. We choose the chip to have five waveguides, and send fixed coherent laser light, described quantum mechanically by coherent states, into the remaining four/three waveguides. We consider five layers of interference regions in a brick-pattern. We created a training and test dataset, each consisting of 250 random samples. We measure the total mean squared error (MSE) loss after the linear layer. To efficiently optimize the MSE during training, we implement a two-stage optimization. First, we optimize the linear layer given the features provided by the quantum circuit. As this is standard linear regression, it is an easily solved convex optimization problem that, in fact, has an analytical solution. Next, we freeze the weights of the linear layer and optimize the coupling parameters of the quantum circuit, simulated using the Gaussian variational approximation, by taking a gradient descent step. In practice, we use SciPy's L-BFGS-B algorithm to perform the gradient descent.

In Fig. 3, we summarize our results. Panel (a) contains a boxplot showing statistical information on the improvement in the test MSE that can be achieved by optimizing a randomly initialized quantum circuit; we do so as a function of the nonlinear interaction strength. We start by noting that a random circuit is the waveguide equivalent of the currently state-of-the-art quantum reservoirs of [20–23]. In particular we observe a more than ten thousand-fold improvement in median MSE for the currently experimentally feasible interaction strength of $U/\gamma = 0.1$ [28]. The necessity of the photonic nonlinearity is also made manifest by the very poor results in the bin $\log_{10} U/\gamma = -\infty$, i.e., $U = 0$. In panel (b), we showcase an example of a circuit for SMTSS characterization at $U/\gamma = 0.1$. Finally, in panel (c), we show several circuit optimization curves, initialized with random circuits, as measured by the test MSE on the two-mode characterization task for $U/\gamma = 0.1$. We easily achieve an order of magnitude improvement. As we didn't change the overall architecture of the PIC, it is natural that the MSE of the two-mode characterization cannot fully match the single-mode MSE. Indeed, two-mode characterization requires the system to disentangle the statistical cross-correlation between two strongly entangled modes, instead of merely identifying single-mode, uncorrelated, statistics. Nevertheless, we obtain perfectly satisfactory values for the MSE.

## 3.2 Single-photon generation

Our setup for quantum state generation is illustrated in Fig. 2. Coherent states (laser fields) are injected into the waveguides and traverse the first nonlinear quantum PIC, thereby generating a multi-mode entangled output state $\rho_{\text{out}}$. Two different figures of merit (FOM) were investigated to

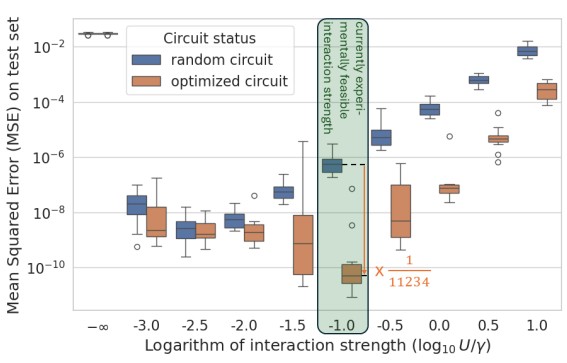

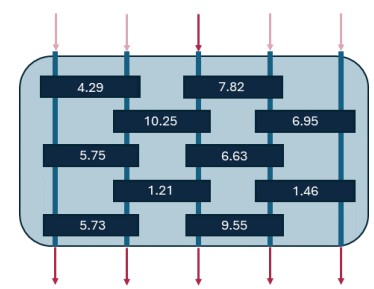

(b) Optimized $J/\gamma$-values for SMTSS at $U/\gamma = 0.1$.

(a) Test mean squared error for SMTSS tomography

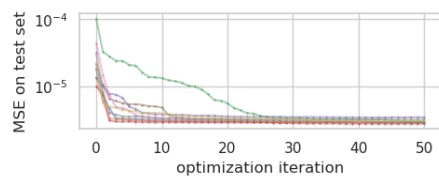

(c) TMTSS tomography optimization curves

Figure 3: Results on single- and two-mode thermal squeezed state (SMTSS and TMTSS) tomography obtained by using a chip as in Fig. 1.

find the optimal configuration of the second, linear, PIC for the task of generating a single-photon output in one output arm, using $\rho_{\text{out}}$ as input. Additional details on the simulation and optimization procedure are provided in the Supplementary Material, see Appendix D.

**Density-density correlations.** The correlation function $g^{(2)}(\tau)$ defines the density-density correlations between a pair of quantum modes $i$ and $j$. It is defined as,

$$g_{ij}^{(2)}(\tau) = \frac{\langle a_i^\dagger(0) a_j^\dagger(\tau) a_j(\tau) a_i(0)\rangle}{n_i(0) n_j(\tau)}. \tag{4}$$

We focus on single-photon generation in one output arm $l$ (i.e., $i = j = l$), at zero time delay ($\tau = 0$). It can be shown that the quantity $g_{ll}^{(2)}(0)$ is non-negative. Moreover, it is only zero for single-photon states. In fact, minimizing $g_{ll}^{(2)}(0; \vec{\theta})$, where $\theta_i$ are the PIC configuration variables, amounts to maximizing the single-photon generation efficiency [32–34].

**Density matrix.** We also formulate a FOM in terms of the entries of the reduced density matrix $\tilde{\rho}_l$ of output arm $l$,

$$\mathcal{F}_l = -\tilde{\rho}_{l,11} - w_0 \tilde{\rho}_{l,00} + \left(\sum_{l,ij} |\tilde{\rho}_{l,ij}| - \tilde{\rho}_{l,11} - \tilde{\rho}_{l,00}\right). \tag{5}$$

Now, the goal is to maximize the matrix entry $\tilde{\rho}_{l,11}$, the single-photon probability, or otherwise filling the vacuum mode $\tilde{\rho}_{l,00}$ ($0 \le w_0 \le 1$). All matrix elements other than $\tilde{\rho}_{l,00}$ and $\tilde{\rho}_{l,11}$ are penalized in the last term. In (5), we *explicitly* aim at maximizing the single-photon probability, which was absent in $g^{(2)}(0)$. Indeed, in (4), very accurate single-photons can be generated, at the cost of obtaining a very low intensity $n_l$.

Nevertheless, measuring $g_{ll}^{(2)}(\tau)$ is a standard task, accomplished in a Hanbury Brown - Twiss interferometric setup (e.g., Refs. [28, 35]), opening the possibility of real-time PIC optimization based on experimental outcomes. In stark contrast, experimentally obtaining matrix entries of $\rho_l$ for (5) is much more complicated [36].

### 3.2.1 Results

Fig. 4 contains an overview of our single-photon generation results. On the left-hand side, in panel (a), a scan of the 2D parameter space $(U, \alpha)$ is shown for moderate interaction strengths $U/\gamma$ and

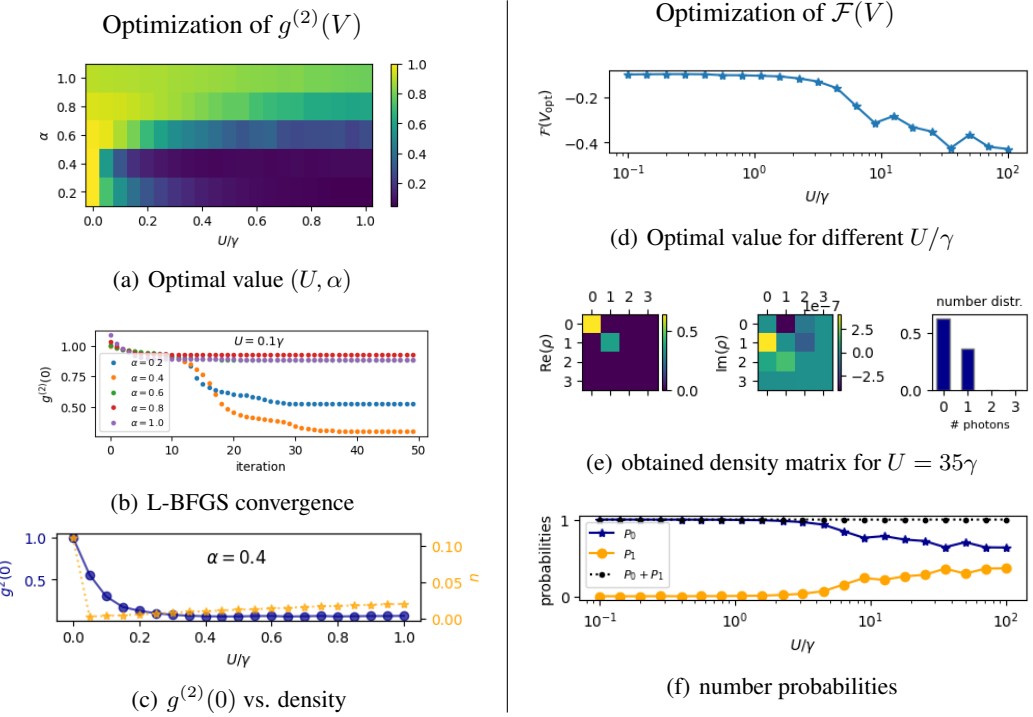

Figure 4: A comparison of the two methods for single-photon state generation using the setup of Fig. 2; (left) the $g^{(2)}(0)$ optimization (4) vs. (right) the matrix-entry objective $\mathcal{F}(V)$ (5).

input laser amplitudes $\alpha$. One notices that the optimal $g^{(2)}(0; V_{\text{opt}})$-values are obtained for large $U$ and small $\alpha$. In panel (b), we illustrate the convergence of PyTorch's L-BFGS optimizer. Noteworthy, as shown in panel (c), while small input amplitudes $\alpha$ and stronger interaction $U$ give good results for $g^{(2)}(0; V_{\text{opt}})$, this comes at the expense of low single-photon output intensities and thus low probabilities – only a few percent chance of single-photon emission, see yellow dots.

On the right-hand side of Fig. 4, we show the results for using the objective $\mathcal{F}(V)$ of Expr. 5. These results were obtained for the same nonlinear PIC configuration, but allowing for larger interaction coefficients $U$. The input intensity is set to $\alpha = 1$. The curve in panel (d) shows some wiggles, but we observe that generally better values of the FOM are obtained at higher $U/\gamma$. In panel (e), the obtained optimal density matrix for $U = 35\gamma$ is shown and it is clear that, to a good approximation, only the matrix entries $\rho_{11}$ and $\rho_{00}$ are occupied. Finally, in panel (f), we compare the probability of producing vacuum and the probability of single-photon generation. We find a single-photon probability of about $p_1 = 0.4$ for large values of $U/\gamma$. Clearly, higher photon numbers (inducing spurious signals ) are strongly suppressed, since $p_0 + p_1 \approx 1$ for all $U$-values (black dotted line).

## 4 Conclusions and Outlook

We have proposed a novel polaritonic PIC to perform universal quantum state characterization and generation. Moreover, we introduced and successfully brought to bear a methodology to optimize the circuit so as to maximize the efficiency of the respective tasks. We provided evidence, based on numerical simulations, of the efficacy of the optimized chips in the tasks of characterizing single- and two-mode squeezed thermal states and the highly accurate generation of single photons.

We are currently actively corresponding with experimental research groups about realizing our proposed PICs, using cryogenic GaAs-based polariton quantum wells. Once implemented, the PIC will be tested and benchmarked against our numerical simulations. Furthermore, for both state tomography and generation, we aim at scaling up our simulations to larger system sizes, using matrix-product based methods [37], to the end of characterizing and generating states of higher complexity – e.g., for performing full density-matrix tomography or generating Bell-pair states. These results will be benchmarked and compared against the original (conceptual) results of Refs. [20, 24]. Finally,

[25] opens up the possibility of enlarging the design space by introducing tunable nonlinearities $U$, while [38] introduces techniques to perform noise-adaptive co-search.

## Acknowledgments and Disclosure of Funding

We would like to thank the inverse design team at HPE labs' Large Scale Integrated Photonics (LSIP) lab (Sean Hooten, Marco Fiorentino, Jared Hulme, Yiwei Peng) and our partners in the project "Quantum Optical Networks based on Exciton-polaritons" (Q-ONE). This project has received funding from the European Union's EIC Horizon Europe Pathfinder Challenges action under the grant agreement No 101115575 (Q-ONE). Views and opinions expressed are however those of the authors only and do not necessarily reflect those of the European Union or European Innovation Council and SMEs Executive Agency (EISMEA). Neither the European Union nor the granting authority can be held responsible for them.

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

# A  Choice of units

In Eqs. (1) and (2) we briefly indicated that we choose to set $\hbar = v_g = 1$, with $\hbar$ the reduced Planck constant and $v_g$ the polariton propagation group velocity. Here we give an overview of the motivation and the conversion to actual experimental units.

A photon with frequency $\omega$ has an energy equal to $E = \hbar\omega$. Thus, by choosing $\hbar = 1$, energy and frequency scales have, by construction, the same units. One can similarly argue for the effect of setting the polariton group velocity to one by considering $d = v_g t$, the distance a polariton propagates, in a time $t$, through the waveguide after injection. The value for $v_g$ can be determined from the waveguide polariton dispersion $\epsilon(k)$, $v_g = \frac{\partial\epsilon(k)}{\partial k}$. Note that it is strongly dependent on the angle of laser incidence and the *Rabi coupling*, the coupling between the photon and exciton field – see Ref. [9].

Unifying the dimensions of time and length, using $v_g$ as the conversion factor, facilitates the simulations but has, on the other hand, a few important assumptions; all photon backscattering, due to roughness in the waveguides, is neglected, as well as potential inter-waveguide dephasing effects, caused by small length mismatches – see the *paraxial* approximation [27].

Our simulations are generally performed with polariton decay rate set to unity, $\gamma = 1$. Consequently, experimentally obtaining a value for polariton lifetime of, e.g., $\tau = \gamma^{-1} = 10\,\mathrm{ps}$ (ps: picoseconds) and for group velocity $v_g = 20\,\mu\mathrm{m}\cdot\mathrm{ps}^{-1}$ allows for the conversion to actual energy, frequency and length units, using $\hbar = 0.658\,\mathrm{meV}\cdot\mathrm{ps}$ (meV: mili-electronvolt). As an example, a value $\frac{U}{\gamma} = 0.1$ from the main text, would, in this case, correspond to $U = 0.1\cdot\hbar\gamma = 66\mu\mathrm{eV}$ and a circuit length of $T = 0.5\,\gamma^{-1}$ to $T = 0.5\cdot v_g\tau = 100\mu m$.

# B  Quantum simulation of the polariton circuit dynamics

Simulating the dynamics of a quantum system is achieved by integrating the Schrödinger equation,

$$i\partial_t|\psi(t)\rangle = H(t)|\psi(t)\rangle. \tag{6}$$

Here, $|\psi(t)\rangle$ is conveniently represented as the quantum-state vector, $|\psi\rangle = \vec{\psi}$, where each vector element $\psi_{\vec{n}}$ is a complex numbers (amplitude and phase). Evaluating $p_{\vec{n}} = |\psi_{\vec{n}}|^2$ gives the probability of detecting the system in a number state $\vec{n}$, upon performing a photon-resolved measurement. The number states are labeled as $\vec{n} = \left[n_1, n_2, \ldots, n_L\right]^T$, with $n_i$ the number of photons in waveguide $i$ and we consider $L$ waveguides in total. The core obstacle of quantum simulation on classical hardware is the exponential increase of the vector space (Hilbert space). Indeed, suppose we want to track up to $N$ photons in one channel, and we have $L$ channels in total, the full vector-space dimension is given by $D = N^L$.

The Hamiltonian $H(t)$ appearing in Eq. 6 is the energy operator applied to the quantum state $|\psi\rangle$. It is represented as a $D \times D$ matrix, acting on the state vector $\psi_{\vec{n}}$. For the photons traversing a set of $L$ coupled nonlinear waveguides, the space axis of photon propagation can be mapped to an effective time dimension, for which the Hamiltonian is given by,

$$H(t) = -\Delta\sum_{i=1}^{L} a_i^\dagger a_i - \sum_{\langle i,j\rangle}\left(J_{ij}(t)a_i^\dagger a_j + J_{ij}^*(t)a_i a_j^\dagger\right) + \frac{U}{2}\sum_{i=1}^{L} a_i^\dagger a_i^\dagger a_i a_i. \tag{7}$$

Here, $a_i$ and $a_i^\dagger$ are, respectively, the bosonic annihilation and creation operators. The former annihilates one photon from waveguide $i$ with $n_i$ photons, $a_i|n_i\rangle = \sqrt{n_i}|n_i - 1\rangle$, the latter injects a photon, $a_i^\dagger|n_i\rangle = \sqrt{n_i + 1}|n_i + 1\rangle$, and their combination gives the photon number, $a_i^\dagger a_i|n_i\rangle = n_i|n_i\rangle$.

The parameter $\Delta = \omega_L - \omega_{LP}$ is the detuning between frequency of the incoming light $\omega_L$ and the lower-polariton resonance $\omega_{LP}$. The matrix $J(t)$ is the graph matrix of waveguide couplings, so each matrix entry $J_{ij}(t)$ describes the tunneling amplitude of one photon from waveguide $i$ to waveguide $j$ at time $t$. The parameters $J_{ij}(t)$ are kept explicitly time-dependent (or, equivalently, distance-dependent in the paraxial approximation), since, by design, couplings between waveguides change as photons cross different layers imprinted on the chip. Finally, the parameter $U$ is constant and quantifies the photonic nonlinearity.

Integrating the Schrödinger equation (6) assumes coherent energy-conserving dynamics in a perfectly isolated environment, without any effects of dissipation or noise. However, in polaritonic systems, polariton losses (due to exciton non-radiative decay, photon scattering...) are unavoidable and must be included in the dynamics. This is described by the the Lindblad Master Equation [26],

$$\partial_t \rho = \mathcal{L}\rho \equiv -\frac{i}{\hbar}\big[H(t), \rho\big] + \gamma \sum_i \Big(a_i \rho a_i^\dagger - \frac{1}{2}\big(a_i^\dagger a_i \rho + \rho a_i^\dagger a_i\big)\Big), \tag{8}$$

in which the first term describes the coherent unitary dynamics, analogous to Eq. (6), and the second the photonic losses in the waveguides (operators $a_i$) with a (uniform) rate $\gamma$. Often this is described by the Liouvillian $\mathcal{L}$, a *superoperator* of which the spectrum shows interesting details about the dissipative dynamics and convergence to the steady state [39].

Crucially, now the quantum simulation must be evaluated on the full *density matrix* $\rho$, which is a $D \times D$ matrix that can be composed in a set of pure states,

$$\rho \equiv \sum_k p_k |\psi_k\rangle\langle|\psi_k|, \tag{9}$$

with $p_k$ the probability of encountering the pure quantum state $k$. Importantly, the representation in terms of a quantum state ensemble $|\psi_k\rangle$ is not unique and this relates back to a fundamental invariance of the Lindblad master equation (8) [40, 41].

Fortunately, the full simulation of the master equation can be circumvented by using a Monte Carlo (MC) sampling technique to collect a set of $k$ pure quantum states $|\psi_k\rangle$ (so $D$-dimensional, not $D^2$), which approximately describe the density matrix in some *unraveling*,

$$\rho(t) \approx \frac{1}{N_s} \sum_k |\psi_k(t)\rangle\langle\psi_k(t)|. \tag{10}$$

Importantly, the density must never be explicitly evaluated from the sampled states $|\psi_k\rangle$ if only expectation values of operators are important (that is, measurement outcomes),

$$\langle O \rangle = \frac{1}{N_s} \sum_k \langle\psi_k|O|\psi_k\rangle \tag{11}$$

The method was originally introduced, independently, in Refs. [29, 42], see Ref. [26] for a good overview. We resorted to the MC quantum trajectory technique for the quantum simulation needed for state generation in Sec. 3.2.

Numerically, the quantum simulations were performed using the python package qutip [43].

## C The Gaussian variational approximation of the dynamics

Rather than performing a full quantum simulation of the Lindblad master equation (8), the dynamics can be captured using the *ansatz* that the quantum states attain Gaussian statistics.

In quantum mechanics, a Gaussian quantum state obeys Gaussian number statistics and is, as a direct consequence, fully defined by the following quantities:

- Operator expectation value: $\alpha_i := \langle a_i \rangle$,

- Normal operator correlations: $\mathcal{N}_{ij} := \langle \delta a_i^\dagger \delta a_j \rangle$,

- Anomalous operator correlations: $\mathcal{C}_{ij} := \langle \delta a_i \delta a_j \rangle$.

Above, we defined the fluctuation operator $\delta a_i := a_i - \langle a_i \rangle$.

A coupled set of closed-form nonlinear ordinary differential equations (ODEs) can be derived to describe the lossy circuit dynamics, within the approximation of the photons obeying Gaussian number statistics – the so-called Hartree-Fock-Bogoliubov method (see, e.g., Ref. [44] for a clean derivation),

$$i\partial_t\alpha_i = \left(-\Delta - i\frac{\gamma}{2} + U|\alpha_i|^2\right)\alpha_i - J_{i,i+1}\alpha_{i+1} - J_{i,i-1}^*\alpha_{i-1} + U\left(2n_i\alpha_i + c_i\alpha_i^*\right) \quad (12)$$

$$i\partial_t\mathcal{N}_{ij} = -i\gamma\mathcal{N}_{ij} + J_{i+1,i}\mathcal{N}_{i+1,j} + J_{i-1,i}^*\mathcal{N}_{i-1,j} - J_{j,j+1}\mathcal{N}_{i,j+1} - J_{j,j-1}^*\mathcal{N}_{i,j-1} \quad (13)$$

$$+ U\left(2\mathcal{N}_{ij}\left(|\alpha_j|^2 + n_j - |\alpha_i|^2 - n_i\right) + \mathcal{C}_{ij}^*\left(\alpha_j^2 + c_j\right) - \mathcal{C}_{ij}\left(\alpha_i^{*2} + c_i^*\right)\right)$$

$$i\partial_t\mathcal{C}_{ij} = -(2\Delta + i\gamma)\mathcal{C}_{ij} - J_{i+1,i}\mathcal{C}_{i+1,j} - J_{i-1,i}^*\mathcal{C}_{i-1,j} - J_{j,j+1}\mathcal{C}_{i,j+1} - J_{j,j-1}^*\mathcal{C}_{i,j-1} \quad (14)$$

$$+ U\left(2\mathcal{C}_{ij}\left(|\alpha_i|^2 + |\alpha_j|^2 + n_i + n_j\right) + \frac{1}{2}\left(\alpha_i^2 + c_i\right)\left(2\mathcal{N}_{ij} + \delta_{ij}\right)\right.$$

$$\left.+ \frac{1}{2}\left(\alpha_j^2 + c_j\right)\left(2\mathcal{N}_{ij}^* + \delta_{ij}\right)\right) \quad (15)$$

Here, we defined $n_i := \mathcal{N}_{ii}$ and $c_i := \mathcal{C}_{ii}$ and only nearest-neighbor couplings $J_{i,i\pm1}$ are considered.

Using standard methods (e.g., Runge-Kutta-45), the system of equations (12)-(14) can be integrated to simulate the Gaussian quantum state of photons emitted by the coupled nonlinear waveguides. Evaluating the waveguide intensities at time $t$ is easily done by reading out $\mathcal{I}_i(t) := |\alpha_i(t)|^2 + n_i(t)$.

Notably, assuming Gaussian statistics of the photonic quantum state greatly reduces the complexity of quantum simulation. Indeed, instead of requiring the quantum simulation of the Schrödinger equation (coherent) (6) or Lindblad master equation (8) in the full exponential Hilbert space with dimension $D = N^L$, we can now resort to a set of coupled nonlinear ODEs (12)-(14) – with polynomial scaling, *not* exponential. Of course, this gain in efficiency comes at the cost of omitting part of the quantum correlations that builds up during quantum time evolution. For short enough times and sufficiently low nonlinearity $U$, however, we see the results can be reliable [44].

## D    Optimizing the PIC for single-photon generation

The task of designing a PIC to obtain an optimal train of single photons as output consists of two parts, as was depicted in the main text in Fig. 2. First, coherent laser light is coupled into a first non-tunable PIC, with a polaritonic nonlinearity, to generate a multimode entangled output state. This generated state is next injected in a fully linear PIC for the single-photon optimization task. We describe in detail the simulations below.

### D.1    Quantum simulation of the lossy circuit

The input laser light is defined as a coherent state $|\beta\rangle$, which have as defining property that they are eigenstates of the bosonic annihilation operator, $a|\beta\rangle = \beta|\beta\rangle$. As a direct consequence, coherent states show Poissonian number statistics, $\langle a^\dagger a\rangle = \text{Var}(a^\dagger a) = |\beta|^2$, with $n = \langle a^\dagger a\rangle$ the number expectation value. A single-frequency continuous wave (CW) – i.e., alaser beam – is described by a coherent state $|\beta\rangle$, where the complex number $\beta = |\beta|e^{i\theta}$ defines the amplitude and phase of the laser.

The output light of the first circuit is represented by the density matrix, expressed using the sampled MC trajectory states,

$$\rho_{\text{out}} = \frac{1}{N_s}\sum_{s=1}^{N_s}|\psi_s\rangle\langle\psi_s|. \quad (16)$$

For a given optical quantum circuit, characterized by the interaction strength $U$, the dissipation rate $\gamma$ and a set of waveguide couplings $J_{ij}$ (see Eqs. (2) and (1)), a set of sampled quantum states $|\psi_s\rangle$ was stored on local hardware for the ensuing task of optimizing the linear circuit.

### D.2    Optimizing the Linear PIC for single-photon generation

The generated set of MC quantum states $|\psi_k\rangle$, describing the density matrix $\rho_{\text{out}}$, are now injected in a second, fully linear PIC. We aim at maximizing the probability of generating a single-photon as output in one of the output channels of this second PIC. Given that (i) the PIC is now fully linear and (ii) we neglect losses during optimization, we can circumvent a cumbersome full quantum simulation

of its dynamics and resort to an equivalent operator formalism – see [45]. The output modes of the PIC are now a linear transform of the original bosonic operators,

$$b_i(V) = \sum_j V_{ij} a_j, \tag{17}$$

with $V$ the unitary matrix to be optimized and representing the PIC. $V$ is of size $L \times L$ (so *quadratic* in the number of waveguides, *not* exponential). We define it as the matrix exponent of a Hermitian matrix $\mathcal{G}$ that represents the graph of waveguide connections,

$$V(\vec{\theta}) := e^{i\mathcal{G}(\vec{\theta})}. \tag{18}$$

Since the only restriction to $\mathcal{G}$ is it being Hermitian, this guarantees a straightforward and efficient expression of $V$ in terms of optimization variables $\theta_i$.

Expressing the optimization objective in terms of the transformed operators $b_i(V)$ from Eq. (17) is essential. For single-photon generation, we present two possible objectives,

**The density-density correlations.** The coincidence rate of photons is quantified by the $g^{(2)}(0)$ function, defined as,

$$g^{(2)}_{ll}(V) = \frac{\langle b_l^\dagger(V) b_l^\dagger(V) b_l(V) b_l(V) \rangle}{n_l^2(V)}, \tag{19}$$

where $\langle \cdot \rangle$ is the *expectation value* of a quantum state $\rho$, here evaluated over the MC sampled trajectory states (see Eq. (11)). We optimize for one output mode $l$. Essentially, (19) represents the ratio of fourth-order correlations ($\sim$ density variation) over density squared. For a coherent laser beam, thanks to the Poissonian number statistics, $g^{(2)}_l(V) = 1$, regardless of the choice of $V$. In contrast, $g^{(2)}_l(V) = 0$ is the correlation value of a single-photon state. Thus, we want to minimize Expr. (19), to approach a single-photon state as closely as possible. Any value $0 \le g^{(2)}_l(V) < 1$ is considered a genuine signature of quantum statistics, referred to as *antibunched* statistics ($\sim$ the particle nature of light)

**Density matrix entries.** A considerable problem with the optimization of $g^{(2)}_l(V) = 0$ is that optimal values are typically obtained at very low photon density (intensity) $n_l$ – there is no *a priori* constraint for ensuring high $n_l$ .

Therefore, besides optimizing the accuracy of a single-photon outcome, we also aim at maximizing the probability for this to occur. We propose an objective expressed in terms of density-matrix entries of the density matrix $\tilde{\rho}$ of output mode $l$,

$$\mathcal{F}(V) = -\tilde{\rho}_{11}(V) - w_0 \tilde{\rho}_{00}(V) + \Big( \sum_{ij} |\tilde{\rho}_{ij}(V)| - \tilde{\rho}_{11}(V) - \tilde{\rho}_{00}(V) \Big). \tag{20}$$

That is, we want to maximize the single-photon matrix entry $\tilde{\rho}_{11}$ and, with a lower weight ($0 < w_0 < 1$), the vacuum entry $\tilde{\rho}_{00}$. The occupation of any other matrix entry is penalized (last term). Note that a non-zero value vacuum entry $\rho_{00}$ is *not* penalized in (20). The relatively low input intensities, $\alpha_i \le 1$, necessitate a non-zero occupation of the vacuum state. Moreover, if no single photon is emitted, we prefer to have a clear-cut zero signal, instead of generating spurious signals with higher photon numbers.

The matrix entries are expressed using the transformed operators from (17),

$$\tilde{\rho}_{mn}(V) = C \cdot \frac{\langle 0| b_l^m(V) \times \rho_{\text{out}} \times b_l^{\dagger n}(V) |0 \rangle}{\sqrt{m!n!}}, \tag{21}$$

with $\rho_{\text{out}}$ the density matrix representing the output state of the first, nonlinear optical network. $C$ is some constant of normalization – diagonal entries of $\tilde{\rho}$ are probabilities, so tr$\tilde{\rho} = 1$ by definition.

Both objectives are conveniently computed as tensor contractions. For this, the stored MC quantum states $|\psi_s\rangle$ are loaded and appropriately contracted with the PIC unitary $V$, each iteration in the

optimization. As an example, obtaining the fourth-order correlation from Expr. (19) requires the evaluation of,

$$\left\langle b_l^\dagger(V)b_l^\dagger(V)b_l(V)b_l(V)\right\rangle = \frac{1}{N_s}\sum_{s=1}^{N_s}\sum_{i_1,i_2,i_3,i_4=1}^{L}V_{l,i_1}^*V_{l,i_2}^*V_{l,i_3}V_{l,i_4}\left\langle a_{i_1}^\dagger a_{i_2}^\dagger a_{i_3}a_{i_4}\right\rangle_s. \quad (22)$$

The fourth-order correlator can be precomputed and stored as a rank-4 tensor, to be reused for every optimization step,

$$A_{i_1,i_2,i_3,i_4} := \frac{1}{N_s}\sum_{s=1}^{N_s}\langle a_{i_1}^\dagger a_{i_2}^\dagger a_{i_3}a_{i_4}\rangle_s. \quad (23)$$

Starting from the generated set of quantum Monte Carlo trajectory states $|\psi_k\rangle$, the PIC unitary $V$ was optimized using PyTorch [46] with the LBFGS solver.

## E   Simulation parameters

**Quantum Tomography.**   The single/two-mode squeezed thermal state is injected into the third (and fourth) waveguide out of the five waveguides of the PIC. Into the remaining four/three waveguides, we inject coherent states with displacement amplitudes and phases, in order, 0.5, 0.1, (0.1,) 0.5 and $-\pi/2$, $-\pi/4$, $(\pi/4,)$ $\pi/2$ respectively. We consider five layers of interference regions in a brick-pattern, each of duration $\Delta t = 0.1/\gamma$, and initialize the hopping coefficients randomly in the range $[0.5, 1.5] \times \frac{\pi}{4\Delta t}$. Note that, in the linear regime, $J = \frac{\pi}{4\Delta t}$ corresponds to a 50:50 beamsplitter. We created a training and test dataset of single- and two-mode thermal squeezed states, each consisting of 250 samples, by randomly sampling in the intervals $r \in [0, 0.4], \theta \in [-\pi, \pi], \bar{n} \in [0, 0.2]$. Finally, the optimization domain of each of the coupling coefficients $J_{ij}$ is bounded to the range $[0, 2\frac{\pi}{4\Delta t}]$.

**Single-photon generation.**   We simulated a first nonlinear PIC consisting of 5 waveguides and 10 stacked, brick-pattern layers. The PIC $J_{ij}(t)$-values were presampled (constant for each $(\alpha, U)$ datapoint in Fig. 2(a)) in the range $[0.5, 1.5] \times \frac{\pi}{4\Delta t}$, with $\Delta t$ the photon propagation time through one layer. The 5 laser input fields were chosen with uniform amplitude $\alpha$, but with different phases, $[-\frac{\pi}{2}, -\frac{\pi}{4}, 0, \frac{\pi}{4}, \frac{\pi}{2}]$. The total circuit duration is $T = \frac{0.5}{\gamma}$. We collected $N_s = 500$ MC quantum samples for each datapoint.

For the optimization task for single-photon generation in the second linear PIC, we used the LBFGS solver of the PyTorch library [46]. The graph matrix $\mathcal{G}$ was represented in the basis of trace zero Hermitian matrices, so that $V$ is a unitary from the special unitary group $\mathrm{SU}(L)$, having the condition $\det V = 1$. The coefficients of the basis matrices, $N^2 - 1$ real numbers, were chosen randomly from the normal distribution (so $\mu = 0$ and $\sigma^2 = 1$) for the initial value. Each datapoint $(\alpha, U)$ started from the same presampled random condition to guarantee consistency for comparison.

