# OpenReview forum: "Designing Polaritonic Integrated Circuits for Quantum Processing"
_NeurIPS.cc/2024/Workshop/MLNCP — MLNCP Poster_

### Official Review · Reviewer_GAjB · 2024-10-01
**Timely study on the role of polaritons in quantum architectures optimized using ML**

**Rating:** 7
**Confidence:** 4

**Review:**

I have read the paper "Designing Polaritonic Integrated Circuits for Quantum Processing" with interest. This is a well written study with a  clear message on the potential of polaritonic circuitry for quantum information processing enhanced by ML techniques (extreme learning techniques). The timing is quite appropriate since a recent report from the Rapaport group has shown impressive progress in using dipolaritons in waveguides for enhanced nonlinearities, https://journals.aps.org/prx/abstract/10.1103/PhysRevX.14.031022 , also there are several EU funded projects aimed towards bringing polaritons closer towards neuromorphic platforms [e.g. EIC Pathfinder Open – 2023 „QUONDENSATE” ; EIC Pathfinder Open – 2023 „PolArt” ; EIC Pathfinder Challenges – 2022 „Q-One”]. Also, https://doi.org/10.1021/acs.nanolett.0c04696 is a relevant reference (albeit classical polaritonics) which the authors might benefit from.

From a modeling perspective, the equations are quite well known to a quantum optics system with a local chi-3 nonlinear term and losses (Lindblad master equation approach). None-the-less, I think the study is nice because the authors are not pumping the system aggressively and creating a lot of dephasing exciton background, which permits them to work with standard quantum optical Hamiltonian with interactions. But I am curious why the authors don't mention the electric field (gates) which is important to get dipolaritons (see PRX). Would electrical contacts be a problem from a design perspective in the context of directional couplers and phase shifters?

The authors study has the word "polariton" in the title, but I struggled to understand ... why. It seems that the only "polariton" aspect of the study is the choice of parameter U/gamma = 0.1 based on recent experiments. I wonder if this justifies calling the study "polaritonic" without explaining in more details what are the materials needed to achieve strong light-matter coupling, is it a cryogenic setup to stabilize the excitons or is it room-temperature, etc. Otherwise it feels like the study is about arbitrary lossy quantum optical system with waveguides. This is not necessarily a bad thing, but it gives of the feeling that polaritons are being used as a "hook".

Another comment is regarding the training of the structure described in lines 123-129. I got a bit lost there. Do I understand correctly that the nonlinear layer is trained for tomographic purposes, but for quantum state generation it is kept fixed and random? And of course, the linear layers are trained in each case? This could maybe be made a bit more explicit/clear in the text since the schematic figure says that the nonlinear layer is fixed but then the text seems to say otherwise.

The results of the study look excellent. I have no criticism here. It seems that the system performs well and the underlying quantum-optical equations and ML methodology applied is well founded.

Lines 186-187 are rather strange with a statement that the g_2 is always positive and then a following statement syaing that it can be zero (contradictory). It would be better to say that g_2 is strictly "non-negative".

---

### Decision · Program_Chairs · 2024-10-10

Accept (Poster)